# Disorder of Golgi Apparatus Precedes Anoxia-Induced Pathology of Mitochondria

**DOI:** 10.3390/ijms24054432

**Published:** 2023-02-23

**Authors:** Yury M. Morozov, Pasko Rakic

**Affiliations:** Department of Neuroscience, Kavli Institute for Neuroscience, Yale School of Medicine, Yale University, New Haven, CT 06510, USA

**Keywords:** ultrastructural pathology, mouse embryo brain, electron microscopy 3D reconstruction, SLP2

## Abstract

Mitochondrial malfunction and morphologic disorganization have been observed in brain cells as part of complex pathological changes. However, it is unclear what may be the role of mitochondria in the initiation of pathologic processes or if mitochondrial disorders are consequences of earlier events. We analyzed the morphologic reorganization of organelles in an embryonic mouse brain during acute anoxia using an immunohistochemical identification of the disordered mitochondria, followed by electron microscopic three-dimensional (3D) reconstruction. We found swelling of the mitochondrial matrix after 3 h anoxia and probable dissociation of mitochondrial stomatin-like protein 2 (SLP2)-containing complexes after 4.5 h anoxia in the neocortex, hippocampus, and lateral ganglionic eminence. Surprisingly, deformation of the Golgi apparatus (GA) was detected already after 1 h of anoxia, when the mitochondria and other organelles still had a normal ultrastructure. The disordered GA showed concentrical swirling of the cisternae and formed spherical onion-like structures with the trans-cisterna in the center of the sphere. Such disturbance of the Golgi architecture likely interferes with its function for post-translational protein modification and secretory trafficking. Thus, the GA in embryonic mouse brain cells may be more vulnerable to anoxic conditions than the other organelles, including mitochondria.

## 1. Introduction

In eukaryotic cells, mitochondria serve as the primary platform for cellular energy supply and, in parallel, play crucial roles in other metabolic processes such as calcium and reactive oxygen species homeostasis and amino acid metabolism. Mitochondrial impairment has been demonstrated as an important stage in various types of brain malfunctions, including developmental abnormalities, trauma, and debilitating age-related diseases [1,2,3,4,5,6]. Mitochondrial function is tightly linked to their morphology: healthy mitochondria are normally thin and long, while dysfunctional ones are short and swollen [7,8,9]. Proper cellular function demands a balance between the opposing mitochondrial dynamics—fusion and fission. Fusion may allow mitochondria to compensate for defects by sharing components; while fission segregates the damaged segments of mitochondria, which then undergo mitophagy in the autophagosome, preserving the integrity of the mitochondrial network [2,3,10]. Disbalance of mitochondrial fusion/fission has been reported in neurodegenerative diseases such as Alzheimer’s disease and Parkinson’s disease [11,12]. The specific mitochondrial phenotype called mitochondria-on-a-string (MOAS) shows an intermingling of thin and enlarged segments [13]. The changes in mitochondrial shape and the pattern of immunolabeling indicate that MOAS may arise from dysregulation of the mitochondrial fission machinery and suggest that mitochondrial division has been initiated, producing constricted segments in the mitochondrial body, but it is unable to complete (unfinished fission). The MOAS phenotype is related to a disbalance of mitochondrial dynamics in the brain of aged macaque and Alzheimer’s disease patients, as well as in traumatized and ischemic brains [13,14,15]. The donut-like shape represents another characteristic mitochondrial phenotype, linked with the increased generation of reactive oxygen species in the mitochondria [16]. The correlation of the mitochondrial donut-shaped morphology with increasing age was demonstrated in the axon terminals of the monkey dorsolateral prefrontal cortex, indicating that mitochondrial changes may play a role in age-related cognitive decline [1]. Thus, electron microscopy detection of mitochondrial morphological disorganization is instrumental for the identification and deep analysis of cellular dysfunction.

The Golgi apparatus (GA) is a highly dynamic constitutive organelle of eukaryotic cells that is composed of stacks of flattened cisternae, often interconnected by membrane tubules and organized in a polarized manner. The GA plays a central role in the glycosylation of proteins and lipids, and other post-translational modifications, with subsequent sorting of the molecules and secretory trafficking. GA-related intracellular transport starts from the endoplasmic reticulum exit sites, which can be connected to the cis-Golgi network. Cargos pass through the stacked medial Golgi cisternae and finally exit from the trans-pole of the GA as clathrin-coated vesicles and immature secretory granules. Some cells also contain a tubular trans-Golgi network with clathrin-coated buds [17,18]. In response to stress, alcohol, and treatments with many pharmacological agents, the GA undergoes a kind of disruption, ranging from mild enlargement of the cisternae lumen to critical fragmentation of the GA, characterized by restructuring into small, round, disconnected, and dispersed vesicles [19]. For instance, Alzheimer’s disease results in fragmentation of the GA cisternae in the cerebellar Purkinje cells and in the hypothalamus neurons. This alteration of the GA may be associated with changes in the microtubules, impaired protein trafficking, and dendritic, spinal, and synaptic pathologies [20,21]. GA fragmentation increases with the age of the animal and positively correlates with tau hyperphosphorylation in the mouse brain [22]. Overexpression of the human longest tau isoform in primary cultures of rat astrocytes induces dispersion, fragmentation, and loss of the GA as an early event [23]. In cultured neurons and in transgenic mice, GA fragmentation can be induced by overexpression of wild type and mutant human tau forms. Importantly, tau is implicated in neuronal GA fragmentation that may occur before the formation of neurofibrillary tangles containing hyperphosphorylated tau [22,24]. Overall, the disruption of GA architecture and function has been widely observed in neurodegenerative diseases, cancer, and other pathologies, but whether GA malfunction is causative to the cell degeneration, or if it is a manifestation of those failures, remains uncertain.

Here, we studied the ultrastructural pathology of 13-day-old (E13) mouse embryo brains during acute anoxia at physiological temperature (37 °C). The anoxic conditions, assumed as total oxygen deprivation with excluded blood circulation, were achieved through decapitation of the embryos and placing the heads in artificial cerebrospinal fluid (ACSF) devoid of dissolved oxygen. Electron microscopy and morphometric analyses of mitochondria were performed in the embryonic subventricular zone (SVZ) because of the relative homogeneity of this brain segment. The SVZ mostly contains cell bodies of neuroprogenitors and immature vertically migrating projection neurons; the SVZ also mostly lacks axons and glial cells that may show different morphologic characteristics. We applied electron microscopy paired with 3D reconstruction from serial ultrathin sections [25,26]. We also used our original procedure of immunohistochemical labeling of stomatin-like protein 2 (SLP2), released from dissociated protein complexes, for the identification of disordered mitochondria that occur ubiquitously in hypoxia-exposed mouse brains [7,27]. We investigated the sequence of GA and mitochondrial disruption and evaluated if the detected ultrastructural pathologies of the cells are permanent or may be reversible after reoxygenation.

## 2. Results

### 2.1. Morphology of GA in Mouse Embryo Brains in Anoxic Conditions

As the first systematic morphologic deviation in developing neurons we identified, the GA cisternae swirl producing a spherical, onion-like phenotype. We observed this GA phenotype after 1 h anoxia, while other organelles, including mitochondria, showed normal ultrastructure (Figure 1 and Figure 2A–D). As previously reported by Bouybayoune and colleagues [28], a swirled GA contains the trans-cisterna in the center of the sphere, which likely interferes with the GA function for secretory trafficking. We detected a similar anoxia-induced disturbance of GA architecture in the SVZ from distinct brain segments. Namely, a dramatic increase in the percentage of onion-like GA was detected after 1 h anoxia in the cerebral cortex, hippocampus, and lateral ganglionic eminence (LGE) (Table 1; Figure 3A). In total, 282 out of 364 GA (around 75%) were swirled after 1 h anoxia. After 3 h anoxia, nearly 100% of identified GA in the cerebral cortex, hippocampus, and LGE (142 out of 143) showed the onion-like phenotype. In the brains fixed immediately after decapitation (control samples), we observed 219 normal GA, and only 2 GA from the cerebral cortex that showed the onion-like phenotype. We did not quantify the GA after 4.5 h anoxia because of the general degradation of the cytoplasm and organelles that make up the GA, as well as the endoplasmic reticulum cisternae unidentifiable in many cells.

### 2.2. Morphology of Mitochondria in Anoxic Conditions

The observation of the mouse embryo brains at a low magnification of the electron microscope showed generally normal ultrastructure after 1 h of acute anoxia (Figure 2). The mitochondria of ellipsoid shape (with variations from elongated to spherical shape) dominated, while other morphological types (branching, donut-like, and MOAS) were rare (Table 2). We did not estimate the numbers and morphology of the cristae, but extensive visual analysis suggests their similarity across the control mice and the 1 h anoxia group (Figure 1). The morphometric analysis of the 3D-reconstructed mitochondria in the cerebral cortex SVZ showed a high correlation of their length and diameter across the control and 1 h anoxia groups (Figure 3B; for the length, F = 1.071, *p* = 0.5607; for the diameter, F = 1.024, *p* = 0.8476). We identified dramatic changes in mitochondrial morphology between 1 h and 3 h of anoxia. After 3 h anoxia, the vast majority of mitochondria looked swollen, with the matrices less electron-dense. The length of the mitochondria significantly decreased, and the diameter increased after 3 h of anoxia compared with the 1 h anoxia group (for the length, F = 27.98, *p* < 0.0001; for the diameter, F = 1.962, *p* < 0.0001). A majority of the 3D-reconstructed mitochondria showed the length nearly equal to the diameter, i.e., the mitochondria developed a nearly spherical shape. Branching, donut-like, and MOAS mitochondria were absent after 3 h anoxia and onwards (Table 2). A morphometric analysis revealed further swelling of mitochondria after 4.5 h of anoxia; the length was not significantly different from the 3 h group (F = 1.106, *p* = 0.5411), while the diameter showed a minor statistically significant increase (F = 1.366, *p* = 0.0468). After 4.5 h anoxia, SLP2-immunopositive mitochondria became very numerous, as detected through light and electron microscopy (Figure 2 and our previous publication [7]). This appears to indicate a massive dissociation of protein complexes in the mitochondria (see chapter: Materials and Methods). At this time point, mitochondrial remnants with visible damage to the inner and outer membranes were detectable in all the analyzed brain segments (such mitochondrial remnants were not the subject of the 3D reconstruction and morphometric analysis). Many cells looked electron-transparent, suggesting the general degradation of cytoplasm and organelles that is characteristic for necrotic-type cell death rather than apoptosis [29]. Thus, our light microscopy quantifications of the SLP2-immunopositive mitochondria [7] and electron microscopy analysis with 3D reconstruction (this article) demonstrate that mitochondria preserve their structure during the first hour of anoxia. A majority of mitochondria swell and acquire a nearly spherical shape after 3 h anoxia. After 4.5 h anoxia, we detected a dramatic dissociation of mitochondrial protein complexes, which takes place in parallel with the massive degradation of cytoplasm and other organelles.

### 2.3. Reaction of GA and Mitochondria to Reoxygenation

We investigated if reactions of GA and mitochondria to anoxia may be reversible in the case of reoxygenation. After 1 h anoxia with subsequent returning of the embryo brains to the oxygenated ACSF, we observed a considerable recovery of GA ultrastructure. In the cerebral cortex, only 3 GA among a total of 71 identified showed the onion-like phenotype. Similarly, in the LGE, only 8 out of 50 GA were swirled (Table 1; Figure 4A). Thus, a majority of GA disrupted during the 1 h anoxia reestablished a normal ultrastructure after the oxygen supply returned. Other organelles, including mitochondria, also demonstrated normal ultrastructure, although we did not perform 3D and morphometric analysis after the 1 h anoxia/reoxygenation. After 3 h anoxia and subsequent reoxygenation, we identified in the cerebral cortex 29 GA, 27 of which showed the onion-like phenotype. Analogically, in the LGE, no normal GA were identified, while 29 GA showed the onion-like phenotype (Table 1; Figure 4A). Therefore, reoxygenation did not recover the ultrastructure of GA after 3 h anoxia when the mitochondrial disruption was significant.

A morphometric analysis of mitochondria in the cerebral cortex after 3 h anoxia with subsequent reoxygenation showed a minor statistically significant increase in the average length and decrease in the average diameter of the mitochondria in comparison with the 3 h anoxia group (Figure 4B; for the length, F = 2.385, *p* < 0.0001; for the diameter, F = 2.223, *p* < 0.0001). This is evidence of the general recovery of the mitochondrial morphology. At the same time, the distributions of both the length and the diameter in the reoxygenation group became wider; this indicates heterogeneity in mitochondrial reactions; some mitochondria probably improve morpho-functional characteristics, while others are disrupted further (Figure 4C,D).

Thus, our findings indicate that the GA ultrastructure in mouse embryo brains is very sensitive to oxygen deprivation, which could be reversible if anoxia-induced cell disruption was minor. The mitochondria preserved normal ultrastructure during the first hour of anoxia and possibly recovered their function without harm to other organelles. Swelling of the mitochondrial matrix took place after 3 h anoxia, and an apparent dissociation of mitochondrial SLP2-containing complexes was observed after 4.5 h anoxia. Such major disruptions of mitochondrial ultrastructure were mostly irreversible after reoxygenation and brought the cells to necrotic-type death.

## 3. Discussion

Here, we are reporting evidence that the molecular structure of the GA in mouse embryonic brain cells is more vulnerable to anoxic conditions than the structure of the mitochondria. We observed ultrastructural pathology of the GA after 1 h anoxia, when the mitochondria showed a normal structure. The disrupted GA show a swirling of the cisternae and forming of an onion-like spherical structure with the trans-cisterna in the center of the sphere [28]. Such a phenotype evidently affects the GA function for the secretory trafficking that normally includes the post-translational modification of proteins in the Golgi cisternae, which evolve from cis- to trans-cisternae, and further form the tubular network or separate vesicles transporting lipids and proteins to certain segments of the cytoplasm or outside of the cell [18]. Our knowledge about the mechanisms and consequences of the formation of onion-like GA is limited. Onion-like Golgi stacks are formed in retrovirus-transformed murine erythroleukemia cells in the process of GA recovery shortly after washout of a fungus metabolite, Brefeldin A, that inhibits protein transport through the endoplasmic reticulum [30]. A similar reformation of GA was described in yeast *Saccharomyces cerevisiae* as an early reaction preceding degradation of the Brefeldin A-treated cells [31]. In transgenic mice carrying the prion protein mutation (homolog of the fatal familial insomnia), onion-like GA form in parallel with an abnormal pattern of intracellular prion accumulation in neurons. These mutant mice also show reduced thalamic and cerebellar volumes—reminiscent of human neuropathology—as well as disease phenotypes such as sleep abnormalities, motor ataxia, and impaired spatial working memory [28]. Our finding of an easy procedure to stimulate the onion-like phenotype of GA in situ in central nervous cells may be instrumental in the study of this vital organelle function and its pathology.

According to our data, changes in morphology of the GA in the mouse embryo brain are the first pathological sign in response to anoxia, which precedes the changes in the mitochondria. Although we observed normal ultrastructure of mitochondria after 1 h of acute anoxia, we suggest that oxygen deficit may reduce oxidative phosphorylation and ATP production without a morphologically detectable disruption in the energy production machinery. Then, the ATP discrepancy provokes the observed ultrastructural pathology and dysfunction of the GA. Such dysfunction may be reversible if the oxygen supply returns quickly. In contrast, if the energy deficit continues long enough, the GA malfunction may damage other organelles, including mitochondria. In turn, damaged mitochondria may initiate numerous irreversible reactions, such as reactive oxygen species production, calcium leakage, cytochrome c leakage, cell apoptosis or necrosis, etc. [4,32,33]. Thus, the molecular architecture of mitochondria, at least in mouse embryo brains, is resistant to anoxia for approximately a 1 h period, whereas intracellular transportation and other functions executed by the GA are vulnerable to a shortage of ATP supply and likely participate in further cellular disruption.

To our knowledge, the direct link between GA phenotypes and the ultrastructure of mitochondria have not been reported and deserves deep investigation. We do not know what molecular mechanisms are responsible for GA swirling. Which pathological conditions, besides acute anoxia as described in this paper, may be linked to up-regulation of onion-like GA? If identified, can GA malfunction help us better understand these pathological processes? Can it serve for the diagnosis or potential treatment of human diseases? Future experiments are warranted to address these and many other intriguing questions.

## 4. Materials and Methods

All the animal protocols complied with the National Institutes of Health (USA) guidelines for animal care and use and were approved by the Institutional Animal Care and Use Committee of Yale University (Protocol #2018-10750.A3, approved 25 June 2020). The C57BL6 mice were housed in the vivarium of Yale University on a recommended diet. For the terminal surgery, the animals were deeply anesthetized with euthasol (0.5 mL/kg body weight) or sodium pentobarbital (3 mL/kg). The timed pregnant mice were killed by cervical translocation and the 13-day-old (E13) embryos exteriorized from the uterus, decapitated, and the heads were exposed to anoxia as previously described [7]. Namely, the embryo heads were submerged in artificial cerebrospinal fluid (ACSF) in closed vials and were placed in an incubator at 37 °C for 1 h (N = 7), 3 h (N = 7), or 4.5 h (N = 2). The ACSF contained: 124 mM NaCl, 2.5 mM KCl, 25 mM NaHCO_3_, 1.25 mM NaH_2_PO_4_, 2 mM MgCl_2_, 2 mM CaCl_2_, and 10 mM dextrose. Prior to the embryo heads’ incubation, the ACSF was bubbled with nitrogen in the incubator for 1 h to balance the temperature and remove the dissolved oxygen. Some of the embryo heads, after 1 h (N = 2) and 3 h (N = 3) of acute anoxia, were kept in the ACSF with 95% O_2_/5% CO_2_ bubbling for 2 h at 37 °C. Immediately after the anoxia or reoxygenation, the heads were immersed overnight in a fixative containing 4% paraformaldehyde, 0.2% picric acid, and 0.2% glutaraldehyde. The brains from littermate embryos (N = 4), fixed with the same fixative immediately after decapitation, were used as the control. Coronal 100-μm-thick brain sections were cut with a Leica VT1000S vibratome (Leica Biosystems, Deer Park, IL, USA) and prepared for immunohistochemistry analysis and electron microscopy, as described below.

We applied immunolabelling with polyclonal made-in-guinea pig antibodies against the last 31 amino acids (L31) of mouse cannabinoid type 1 receptor (CB_1_R) from Frontier Science Co., Ltd. (Ishikari, Hokkaido, Japan; catalog numbers CB1-GP-Af530). We previously demonstrated the double specificity of CB_1_R-L31 antibodies that, in parallel with CB_1_R, also bind to mitochondrial stomatin-like protein 2 (SLP2) [7,27,34]. According to the modern data, SLP2 is a mitochondrial inner membrane-associated protein that faces the matrix and is incorporated in a complex with other proteins. SLP2 plays several crucial roles, including maintaining the stability of the mitochondrial function [35,36,37,38,39]. We found that CB_1_R-L31 antibodies label exclusively the swollen mitochondria in mouse embryo brains, providing an important benefit of this serum application for the analysis of disordered mitochondria in situ [7]. We hypothesize that SLP2-containing complexes may be dissociated in certain conditions, making SLP2′s epitope available for immunolabeling (Figure 5). Disordered cells containing swollen immunopositive mitochondria also contain swollen immunonegative ones [7,27]. We suppose that the denaturation of SLP2 destroys the epitope in parallel with the progression of the mitochondrial pathology. Mouse embryo ventricular and subventricular zones (VZ and SVZ) represent a location convenient for analysis of SLP2-immunopositive mitochondria because CB_1_R-expressing cells and axons are virtually absent from these zones [26].

For the visualization of primary antibodies, we applied either immunoperoxidase or immunogold labeling. For the first procedure, the slices after incubation with CB_1_R-L31 antibodies (dilution 1:2000) were extensively washed and immersed in a solution of biotinylated anti-guinea pig IgGs (1:300) and the Elite ABC kit (1:300, all from Vector Laboratories, Burlingame, CA, USA). Ni-intensified 3,3′-diaminobenzidine–4HCl (DAB-Ni) was used as a chromogen. For the immunogold/silver procedure (dilution of CB_1_R-L31 antibodies 1:200), anti-guinea pig secondary serum conjugated with 1 nm gold particles (1:80; Aurion, Wageningen, The Netherlands) were applied. Silver-intensification of the gold particles was performed with Aurion R-Gent SE-LM, according to the manufacturer’s instructions. Afterwards, the sections were post-fixed with OsO_4_, dehydrated, and embedded in Durcupan ACM (Fluka, Buchs, Switzerland) on microscope slides and coverslipped. The brain segments without apparent abnormalities or mechanical damage during the slice preparation were taken for electron microscopy analysis. Selected fragments of tissue were analyzed and photographed with an Axioplan 2 microscope (Zeiss, Jena, Germany) and re-embedded into Durcupan blocks for electron microscopic analysis. The samples were cut with a Reichert Ultracut S ultramicrotome (Leica Biosystems, Deer Park, IL, USA) into 60 nm-thick sections. The sections were then stained with lead citrate and evaluated and photographed in a Talos L120C electron microscope (ThermoFisher Scientific, Boston, MA, USA).

The electron microscopy and morphometric analysis of the mitochondria were performed in the embryonic SVZ because of the relative homogeneity of this brain segment. The SVZ contains mostly cell bodies of neuroprogenitors and immature vertically migrating projection neurons, and mostly lacks axons and glial cells that may show different morphologic characteristics. The embryo SVZ from the neocortex, hippocampus, and lateral ganglionic eminence (LGE) were systematically evaluated with 8500× magnification of the electron microscope, avoiding double observation of the same tissue segments. The straight and onion-like GA were counted, and the percentages of onion-like GA were calculated for each analyzed brain segment from each embryo. The values are expressed as the mean ± SD for each brain segment in the control, anoxic, and reoxygenation conditions.

The 3D reconstructions were performed as we previously described [26], with the following modifications. A series of 40–70 consecutive sections from occasionally taken segments of SVZ from the embryo neocortex were made with 5300× magnification of the electron microscope. A 3D reconstruction of the mitochondria and GA, and measurement of the mitochondria volume, were performed using the computer program Reconstruct 1.1.0.0. (Boston, MA, USA) [40,41], publicly available at http://www.bu.edu/neural/Reconstruct.html (accessed on 17 March 2022). The following morpho-functional types of mitochondria were recognized and counted among the 3D-reconstructed mitochondria: (1) simple elongated mitochondria were identified here as ellipsoid mitochondria, regardless if they are straight or curved (mitochondria of spherical shape were also counted as ellipsoid ones); (2) branched mitochondria that consist of two or more elongated segments—the probable evidence of active fusion; (3) donut-like mitochondria that were reported as the predominant source of reactive oxygen species [16]; (4) mitochondria-on-a-string (MOAS) that show an intermingling of thin and enlarged segments. The MOAS phenotype is related to a disbalance of mitochondria fission/fusion in the brains of aged macaques and a mouse model of Alzheimer’s disease [13,15]. A morphometric characterization of 3D-reconstructed mitochondria was performed, as we validated in our previous article [7]. Namely, the estimated length of the reconstructed mitochondria was calculated as the hypotenuse of a triangle, with one cathetus measured as the maximal horizontal shift of the profiles in the serial sections, and the other cathetus as the number of sections in which the mitochondrial profiles are seen multiplied by the thickness of the sections. The estimated length of the branched mitochondria was calculated as the sum of the lengths of their simple fragments. Truncated mitochondria were treated “as is” in the quantification if their estimated length was more than the average length of the non-truncated mitochondria in the same group; if not, they were ignored. The root-mean-square diameter of the reconstructed mitochondria was calculated based upon the volume formula for ellipsoids using the length, as measured above, as the longest diameter of the ellipsoid. Mitochondria from different embryos in the same experimental conditions were combined in one group for calculation of the average and statistical analysis of variances using an F-test in GraphPad Prism Version 9.4.1 (681) (GraphPad Software, LLC, Boston, MA, USA). The values of mitochondrial length and diameter are expressed as the mean ± SD for every embryo, and for the combined groups. A value of *p* < 0.05 was used as the threshold for significance.

## Figures and Tables

**Figure 1 ijms-24-04432-f001:**
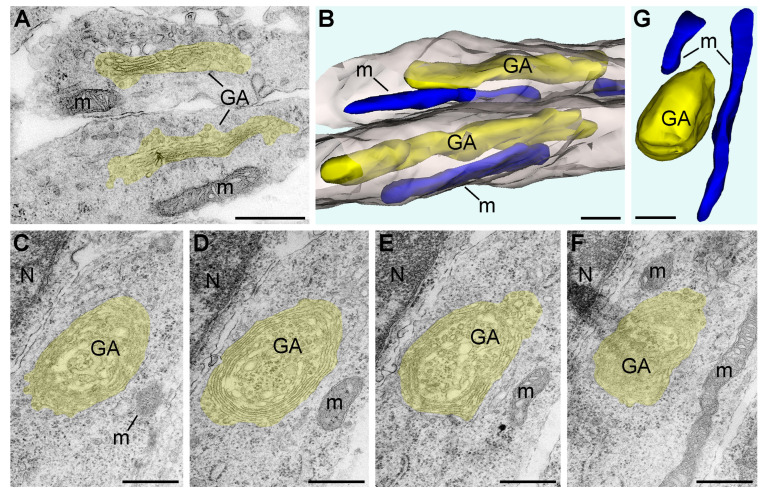
Electron micrographs and 3D reconstructions of GA and mitochondria from E13 mouse embryo cerebral SVZ fixed immediately after decapitation (**A**,**B**) and after 1 h anoxia (**C**–**G**). GA including cisternae and vesicles are shown yellow in the electron micrographs (**A**,**C**–**F**) and the 3D images (**B**,**G**). Mitochondria (m) are depicted blue in the 3D images. Two adjacent processes are displayed as semitransparent grey in the 3D image (**B**). In the control brain, both analyzed GA show normal stacks of cisternae and elongated configurations. Serial micrographs (**C**–**F**) and 3D reconstruction (**G**) demonstrate spherical onion-like shape of GA after 1 h anoxia. Abbreviation: N, cell nucleus. Scale bars, 500 nm.

**Figure 2 ijms-24-04432-f002:**
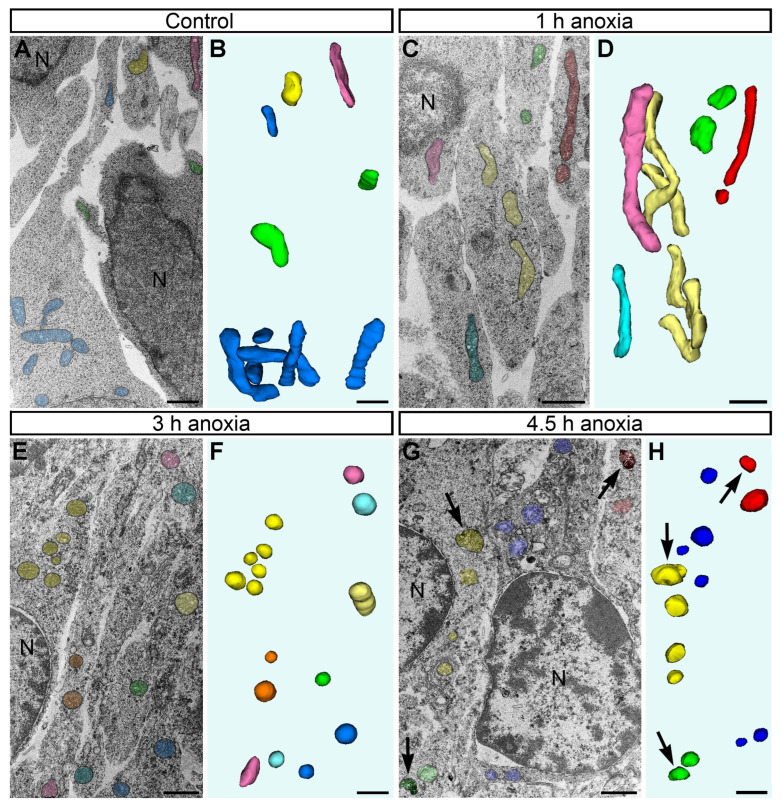
Electron micrographs of neuropil and 3D reconstructions of mitochondria from E13 mouse embryo cerebral SVZ fixed immediately after decapitation (**A**,**B**), after 1 h (**C**,**D**), 3 h (**E**,**F**), and 4.5 h anoxia (**G**,**H**). Mitochondria from distinct cell bodies or processes are shown in same colors in the micrographs and 3D images. Notice that long mitochondria are numerous in control brain and after 1 h anoxia; spherical mitochondria are predominant after 3 and 4.5 h anoxia. SLP2-immunopositive mitochondria (arrows) are numerous after 4.5 h of anoxia. Abbreviation: N, cell nucleus. Scale bars, 1 μm.

**Figure 3 ijms-24-04432-f003:**
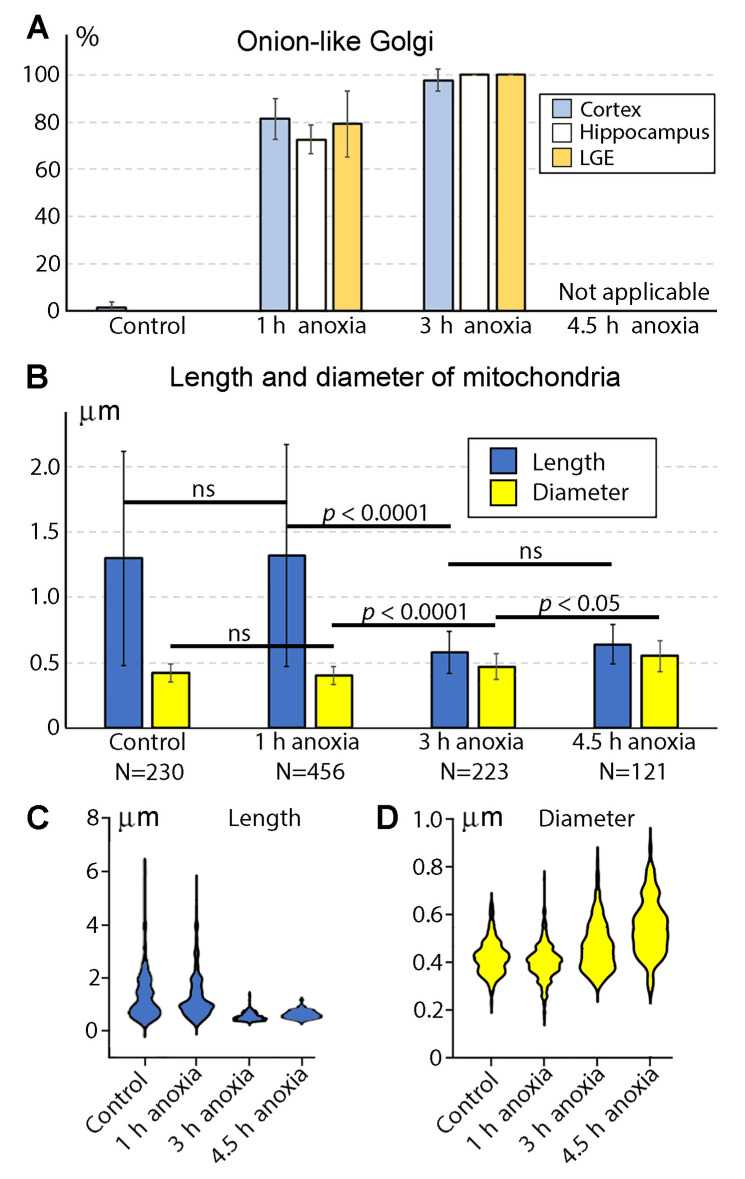
(**A**) Up-regulation of swirled onion-like GA in the anoxia-exposed E13 mouse embryo brains. Swirled GA are nearly absent in control animals. Normal GA are virtually absent after 3 h anoxia. After 4.5 h, even swirled GA are barely identifiable because of degradation of cytoplasm and organelles in many cells; that is why this time point was excluded from analysis. Percentages of swirled GA are expressed as mean ± SD for each spatiotemporal location (Table 1). (**B**) Morphometric characterization of mitochondria in mouse embryo cerebral SVZ fixed immediately after decapitation (control) and after 1, 3, and 4.5 h anoxia. Length and diameter of mitochondria are expressed as mean ± SD. Ns are numbers of analyzed 3D-reconstructed mitochondria. (**C**,**D**) Estimation plots for the length and diameter of mitochondria from control and anoxia-exposed mouse embryo cerebral cortex. Notice similarity between control and 1 h anoxia groups for both length and diameter. Notice also large difference between 3 h and 4.5 h groups from control and 1 h anoxia groups.

**Figure 4 ijms-24-04432-f004:**
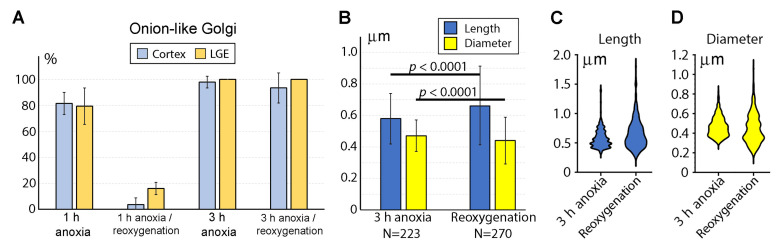
(**A**) Reoxygenation of the anoxia-exposed mouse embryo brains down-regulates percentage of the onion-like GA after 1 h, but not after 3 h anoxia. Percentages of swirled GA are expressed as mean ± SD for each spatiotemporal location (Table 1). (**B**–**D**) Morphometric comparison of mitochondria in the mouse embryo cerebral SVZ after 3 h anoxia versus 3 h anoxia/reoxygenation. (**B**) Length and diameter of mitochondria are expressed as mean ± SD. Ns are numbers of analyzed 3D-reconstructed mitochondria. (**C**,**D**) Estimation plots for length and diameter of the mitochondria.

**Figure 5 ijms-24-04432-f005:**
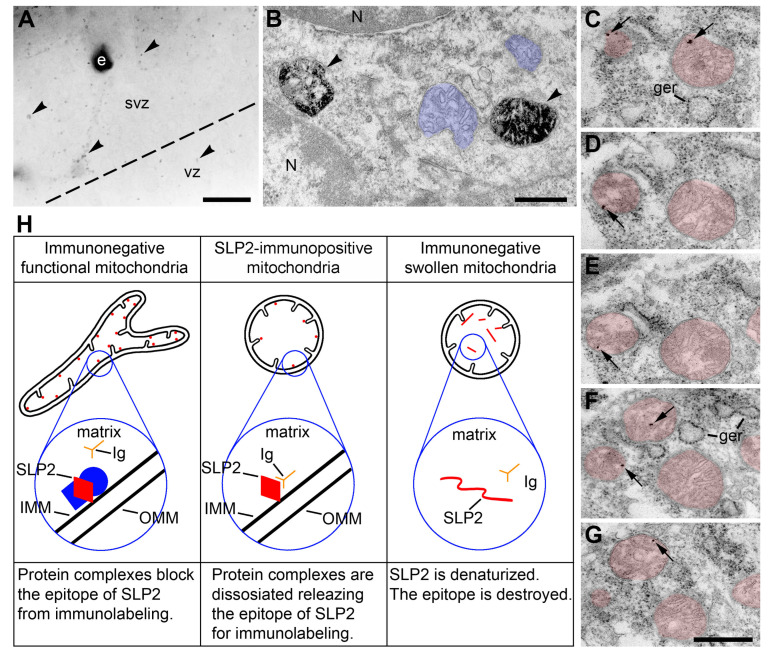
(**A**–**G**) Immunolabeling of disordered mitochondria in the cerebral cortex ventricular zone (vz) and subventricular zone (svz) of E13 mouse embryos after 4.5 h anoxia using CB_1_R-L31 antibodies. (**A**,**B**) Light and electron micrographs of the neuropil labeled with immunoperoxidase-DAB-Ni. Immunopositive mitochondria are pointed to with arrowheads. Notice that black immunoreaction end-product in (**B**) accumulates in the matrix, while the cristae stay immunonegative. Some mitochondria, although swollen, remain immunonegative (highlighted blue). (**C**–G) Serial electron micrographs of mitochondria (highlighted red) immunolabeled with nano-gold and silver amplification. Numerous silver particles (arrows) are located in the mitochondrial matrix and contact the inner membrane in accord with the modern view to location of SLP2 on the inner surface of inner mitochondrial membrane [39]. Scale bars, 20 μm in (**A**); 500 nm in (**B–G**). (**H**) Hypothetical schema of the relationships between mitochondrial morphology, molecular conformation of the SLP2 protein, and CB_1_R-L31 antibodies (Ig). Proteins that make a complex with SLP2 on the inner surface of inner mitochondrial membrane (IMM) are depicted blue. Abbreviations: e, erythrocyte; ger, granular endoplasmic reticulum; N, cell nucleus; OMM, outer mitochondrial membrane.

**Table 1 ijms-24-04432-t001:** Quantification of normal and onion-like GA in E13 mouse embryo brain in control, anoxic, and reoxygenation conditions.

Sample and Embryo #	Normal Golgi		Onion-Like Golgi	
	Number	%	Number	%
Control				
Cortex #1	20	95.24	1	4.76
Cortex #2	26	100.00	0	0.00
Cortex #3	29	96.67	1	3.33
Cortex #4	15	100.00	0	0.00
Cortex total, sum	90		2	
Cortex total, mean ± SD		97.98 ± 2.41		2.02 ± 2.41
Hippocampus #1	9	100.00	0	0.00
Hippocampus #2	14	100.00	0	0.00
Hippocampus #3	28	100.00	0	0.00
Hippocampus #4	19	100.00	0	0.00
Hippocampus total, sum	70		0	
Hippocampus total, mean		100.00		0.00
LGE #1	19	100.00	0	0.00
LGE #2	40	100.00	0	0.00
LGE total, sum	59		0	
LGE total, mean		100.00		0.00
1 h anoxia				
Cortex #1	7	31.82	15	68.18
Cortex #2	2	14.29	12	85.71
Cortex #3	7	22.58	24	77.42
Cortex #4	3	12.00	22	88.00
Cortex #5	1	12.50	7	87.50
Cortex total, sum	20		80	
Cortex total, mean ± SD		18.64 ± 8.51		81.36 ± 8.51
Hippocampus #1	19	31.67	41	68.33
Hippocampus #2	12	23.53	39	76.47
Hippocampus #3	12	33.33	24	66.67
Hippocampus #4	5	20.83	19	79.17
Hippocampus total, sum	48		123	
Hippocampus total, mean ± SD		27.34 ± 6.10		72.66 ± 6.10
LGE #1	10	25.64	29	74.36
LGE #2	7	30.43	16	69.57
LGE #3	7	26.92	19	73.08
LGE #4	0	0.00	15	100.00
LGE total, sum	24		79	
LGE total, mean ± SD		20.75 ± 13.98		79.25 ± 13.98
1 h anoxia/reoxygenation				
Cortex #1	29	100.00	0	0.00
Cortex #2	39	92.86	3	7.14
Cortex total, sum	68		3	
Cortex total, mean ± SD		96.43 ± 5.05		3.57 ± 5.05
LGE #1	21	87.50	3	12.50
LGE #2	21	80.77	5	19.23
LGE total, sum	42		8	
LGE total, mean ± SD		84.13 ± 4.76		15.87 ± 4.76
3 h anoxia				
Cortex #1	0	0.00	4	100.00
Cortex #2	1	9.09	10	90.91
Cortex #3	0	0.00	25	100.00
Cortex #4	0	0.00	18	100.00
Cortex total, sum	1		57	
Cortex total, mean ± SD		2.27 ± 4.55		97.73 ± 4.55
Hippocampus #1	0	0.00	31	100.00
Hippocampus #2	0	0.00	29	100.00
Hippocampus total, sum	0		60	
Hippocampus total, mean		0.00		100.00
LGE #1	0	0.00	14	100.00
LGE #2	0	0.00	11	100.00
LGE total, sum	0		25	
LGE total, mean		0.00		100.00
3 h anoxia/reoxygenation				
Cortex #1	0	0.00	12	100.00
Cortex #2	2	20.00	8	80.00
Cortex #3	0	0.00	7	100.00
Cortex total, sum	2		27	
Cortex total, mean ± SD		6.67 ± 11.55		93.33 ± 11.55
LGE #1	0	0.00	8	100.00
LGE #2	0	0.00	14	100.00
LGE #3	0	0.00	7	100.00
LGE total, sum	0		29	
LGE total, mean		0.00		100.00

**Table 2 ijms-24-04432-t002:** Morphologic characteristics of 3D-reconstructed mitochondria from SVZ of E13 mouse embryo cerebral cortex in control, anoxic, and reoxygenation conditions.

Embryo Analyzed	Ellipsoid	Branched	Donut-Like	MOAS	Total	Mean Length ± SD, µm	Mean Diameter ± SD, µm
Control #1	100	0	5	0	105	1.28 ± 0.71	0.43 ± 0.08
Control #2	53	0	0	2	55	1.20 ± 1.00	0.43 ± 0.05
Control #3	65	0	5	0	70	1.39 ± 0.82	0.40 ± 0.07
Control total	218	0	10	2	230	1.30 ± 0.82	0.42 ± 0.07
1 h anoxia #1	121	2	3	0	126	1.04 ± 0.61	0.40 ± 0.07
1 h anoxia #2	116	1	1	1	119	1.42 ± 0.92	0.38 ± 0.07
1 h anoxia #3	148	1	0	2	151	1.36 ± 0.79	0.41 ± 0.06
1 h anoxia #4	57	2	0	1	60	1.60 ± 1.09	0.44 ± 0.08
1 h total	442	6	4	4	456	1.32 ± 0.85	0.40 ± 0.07
3 h anoxia #1	55	0	0	0	55	0.69 ± 0.21	0.52 ± 0.11
3 h anoxia #2	168	0	0	0	168	0.54 ± 0.12	0.45 ± 0.09
3 h total	223	0	0	0	223	0.58 ± 0.16	0.47 ± 0.10
3 h anoxia/reoxygenation #1	91	0	0	0	91	0.70 ± 0.19	0.57 ± 0.14
3 h anoxia/reoxygenation #2	35	0	0	0	35	0.79 ± 0.32	0.44 ± 0.11
3 h anoxia/reoxygenation #3	141	0	3	0	144	0.59 ± 0.24	0.37 ± 0.10
3 h anoxia/reoxygenation total	267	0	3	0	270	0.66 ± 0.25	0.44 ± 0.15
4.5 h anoxia #1	72	0	0	0	72	0.65 ± 0.17	0.54 ± 0.12
4.5 h anoxia #2	49	0	0	0	49	0.63 ± 0.13	0.56 ± 0.11
4.5 h total	121	0	0	0	121	0.64 ± 0.15	0.55 ± 0.12

## Data Availability

The data presented in this study are available on request from the corresponding author. The data are not publicly available due to absence of a publicly accessible repository.

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
