# Peer review of "Disorder of Golgi Apparatus Precedes Anoxia-Induced Pathology of Mitochondria"

_ijms, 2023, doi:10.3390/ijms24054432_

Round 1
Reviewer 1 Report
The authors present original work describing the response of subcellular organelles, including the Golgi and mitochondria, to acute anoxia. The assessment was done using innovative electron microscopy 3D reconstruction and methodology well established in Dr. Morozov’s previous publications. Overall, technical aspects of the manuscript regarding EM are very good.
The work described in the manuscript could have greater impact on the field if the following questions will be addressed:
1. The merit for selection of time after anoxia (1, 3 and 4.5 hrs) is not clear or well justified. The review of the articles cited in this manuscript revealed that most mitochondrial morphological changes under similar experimental conditions occur relatively early between 1 – 2 hrs (e.g., PMID: 32483593). Thus, it appears that by omitting a 2-hour interval after anoxia the authors missed the transition of long mitochondria to fragmented ones observed at 3 hrs. This is especially unfortunate since this transition most likely involved the formation of MOAS. The recent discovery of MOAS phenotype as a novel and unique phenomenon deserves to be studied in detail. The authors utilized the approach, a 3D EM reconstruction, that is positioned to detect MOAS that can’t be accurately detected using 2D TEM. It still remains unclear how MOAS are formed and whether they are reversible. In this work, this could have been studied but apparently the window of opportunity to observe MOAS was missed. Similar, there were no studies to confirm whether MOAS were reversed to long or small mitochondria after oxygenation (continuous fission or fission arrest). Thus, one of the most interesting and innovative aspects of this study was not interrogated properly.
2. Mitochondria acquire various shapes depending on the compartment and origin of the cell. Mitochondrial morphology differs in dendrites vs axons, in glia and neurons. The authors need to clarify what brain cells and what cellular compartments were taken into investigation. If such information is not available, this should be discussed in Limitations of the Study
3. The authors need to significantly edit the Discussion. It should include better description of the relationship between structure of the Golgi and mitochondria and their functions. Since the authors do not provide any experimental evidence in support of altered functions, the discussion should provide better clarity of these relationships. At the moment, it is unclear how data presented in the manuscript relate to human conditions, what additional information regarding potential treatment options could be gathered.
4. The authors claim that they were investigating the relationship between multiple subcellular organelles and anoxia/reoxygenation. However, only two organelles, mitochondria and the Golgi, were studied. No interrogation of the ER and MERCS were conducted, which is also a missed opportunity. Please correct the statement.
5. The authors should consider including the section “Limitations of the study”.
Author Response
Reviewer 1
The authors present original work describing the response of subcellular organelles, including the Golgi and mitochondria, to acute anoxia. The assessment was done using innovative electron microscopy 3D reconstruction and methodology well established in Dr. Morozov’s previous publications. Overall, technical aspects of the manuscript regarding EM are very good.
The work described in the manuscript could have greater impact on the field if the following questions will be addressed:
- The merit for selection of time after anoxia (1, 3 and 4.5 hrs) is not clear or well justified. The review of the articles cited in this manuscript revealed that most mitochondrial morphological changes under similar experimental conditions occur relatively early between 1 – 2 hrs (e.g., PMID: 32483593). Thus, it appears that by omitting a 2-hour interval after anoxia the authors missed the transition of long mitochondria to fragmented ones observed at 3 hrs. This is especially unfortunate since this transition most likely involved the formation of MOAS. The recent discovery of MOAS phenotype as a novel and unique phenomenon deserves to be studied in detail. The authors utilized the approach, a 3D EM reconstruction, that is positioned to detect MOAS that can’t be accurately detected using 2D TEM. It still remains unclear how MOAS are formed and whether they are reversible. In this work, this could have been studied but apparently the window of opportunity to observe MOAS was missed. Similar, there were no studies to confirm whether MOAS were reversed to long or small mitochondria after oxygenation (continuous fission or fission arrest). Thus, one of the most interesting and innovative aspects of this study was not interrogated properly.
We agree that additional time-points analyzed could provide new data on mitochondrial dynamics in anoxic brain. Unfortunately, time frame provided for the manuscript corrections (10 days) exclude a chance to address this important point. We are grateful to reviewer for this suggestion that we will use in our future studies.
- Mitochondria acquire various shapes depending on the compartment and origin of the cell. Mitochondrial morphology differs in dendrites vs axons, in glia and neurons. The authors need to clarify what brain cells and what cellular compartments were taken into investigation. If such information is not available, this should be discussed in Limitations of the Study
Thank you for this point. We added following sentence in last paragraph of Introduction and in chapter Materials and Methods:
“Electron microscopy and morphometric analyzes of mitochondria were performed in the embryonic SVZ because of relative homogeneity of this brain segment that contains mostly cell bodies of neuroprogenitors and immature vertically migrating projection neurons; SVZ mostly lacks axons and glial cells that may show different morphologic characteristics.”
- The authors need to significantly edit the Discussion. It should include better description of the relationship between structure of the Golgi and mitochondria and their functions. Since the authors do not provide any experimental evidence in support of altered functions, the discussion should provide better clarity of these relationships. At the moment, it is unclear how data presented in the manuscript relate to human conditions, what additional information regarding potential treatment options could be gathered.
Indeed, limited data are available for relationship between GA and mitochondria. We agree that additional analyses would be informative, but such a great task is out of the scope for this article.
We added following sentences in Discussion:
“…if energy deficit continues long enough, GA malfunction may damage other organelles including mitochondria. In turn, damaged mitochondria may initiate numerous irreversible reactions such as reactive oxygen species production, calcium and cytochrome c leakage, cell apoptosis or necrosis, etc (Giacomello et al., 2020; Han et al., 2020; Harland et al., 2020). Thus, molecular architecture of mitochondria, at least in mouse embryo brain, is resistant to anoxia during about 1-hour period, whereas intracellular transportation and other functions executed by GA are vulnerable to shortage of ATP supply and likely participate in further cellular disorder.
To our knowledge, reciprocal influence of GA and mitochondria and, particularly, direct link between GA phenotype and ultrastructure of mitochondria have not been reported and deserve deep investigation. We do not know what molecular mechanisms are responsible for GA swirling. Which pathological conditions, besides acute anoxia as described in this paper, may be linked to up-regulation of onion-like GA? If identified, can GA malfunction help us better understand the pathological processes? Can it serve for diagnosis or potential treatment of human diseases? Future experiments are warranted to address these and many other intriguing questions.”
- The authors claim that they were investigating the relationship between multiple subcellular organelles and anoxia/reoxygenation. However, only two organelles, mitochondria and the Golgi, were studied. No interrogation of the ER and MERCS were conducted, which is also a missed opportunity. Please correct the statement.
We replaced words “other organelles including mitochondria” with “mitochondria” through chapter Discussion.
- The authors should consider including the section “Limitations of the study”.
See reply to point 3 above for the text included in Discussion.
Reviewer 2 Report
Thank you for the opportunity to review this manuscript, dealing with interesting findingsThank you for the opportunity to review this manuscript, dealing with interesting findings entitled “Disorder of Golgi Apparatus Precedes Anoxia-Induced Pathology of Mitochondria”. They analyzed the morphologic reorganization of organelles in the embryonic mouse brain during acute anoxia using immunohistochemical identification of disordered mitochondria followed by electron microscopic 3D reconstruction. They proposed mitochondrial perturbation linked with disturbance of the Golgi architecture likely interferes with its function for post-translational protein modification and secretory trafficking. Overall, their study design is noteworthy, and the results corroborate their hypothesis. However, a few things a confusing in addition to the lack of representation of data. After a few corrections and justification manuscript could be considered for publication. In Figure 1B mitochondria seem apoptotic which reveals the dissociation of mitochondrial stomatin-like protein-2-containing complexes. However, the author confirms it is swollen. Though swollen, apoptotic mitochondria confirm its perturbation. However, the author needs to describe clearly how they confirm morphology. how they confirm mitochondrial swelling If they confirm mitochondrial swelling. The author should provide the best representative Figure 1C1-C5. Images do not properly confirm significant SLP2 protein and CB1R-L31 antibodies complex accumulation in the mitochondrial matrix however, accumulation is higher at the cellular level. It would be nice if the author replaced the bar graph (Figure 4 A, B; Figure 5A, B) with a scatter graph. Since the error bar is larger it would be nice to see the individual value. If the author found any changes in mitochondrial morphology between the hippocampus and cerebral cortex it would be nice if they were addressed in the discussion section. Representative images would be a plus.
entitled “Disorder of Golgi Apparatus Precedes Anoxia-Induced Pathology of Mitochondria”. They analyzed the morphologic reorganization of organelles in the embryonic mouse brain during acute anoxia using immunohistochemical identification of disordered mitochondria followed by electron microscopic 3D reconstruction. They proposed mitochondrial perturbation linked with disturbance of the Golgi architecture likely interferes with its function for post-translational protein modification and secretory trafficking. Overall, their study design is noteworthy, and the results corroborate their hypothesis. However, a few things a confusing in addition to the lack of representation of data. After a few corrections and justification manuscript could be considered for publication. In Figure 1B mitochondria seem apoptotic which reveals the dissociation of mitochondrial stomatin-like protein-2-containing complexes. However, the author confirms it is swollen. Though swollen, apoptotic mitochondria confirm its perturbation. However, the author needs to describe clearly how they confirm morphology. how they confirm mitochondrial swelling If they confirm mitochondrial swelling. The author should provide the best representative Figure 1C1-C5. Images do not properly confirm significant SLP2 protein and CB1R-L31 antibodies complex accumulation in the mitochondrial matrix however, accumulation is higher at the cellular level. It would be nice if the author replaced the bar graph (Figure 4 A, B; Figure 5A, B) with a scatter graph. Since the error bar is larger it would be nice to see the individual value. If the author found any changes in mitochondrial morphology between the hippocampus and cerebral cortex it would be nice if they were addressed in the discussion section. Representative images would be a plus.

Author Response
Reviewer 2
entitled “Disorder of Golgi Apparatus Precedes Anoxia-Induced Pathology of Mitochondria”. They analyzed the morphologic reorganization of organelles in the embryonic mouse brain during acute anoxia using immunohistochemical identification of disordered mitochondria followed by electron microscopic 3D reconstruction. They proposed mitochondrial perturbation linked with disturbance of the Golgi architecture likely interferes with its function for post-translational protein modification and secretory trafficking. Overall, their study design is noteworthy, and the results corroborate their hypothesis. However, a few things a confusing in addition to the lack of representation of data. After a few corrections and justification manuscript could be considered for publication.
In Figure 1B mitochondria seem apoptotic which reveals the dissociation of mitochondrial stomatin-like protein-2-containing complexes. However, the author confirms it is swollen. Though swollen, apoptotic mitochondria confirm its perturbation. However, the author needs to describe clearly how they confirm morphology. how they confirm mitochondrial swelling If they confirm mitochondrial swelling.
This point of criticism is not totally clear for us. We identified swelling of mitochondria by those electron-transparent matrix (if DAB labeling was not applied) visible in single sections and morphometric analysis of 3D reconstructed mitochondria that acquire almost spherical shape. Mitochondrial swelling often observed during many different cell pathologies including apoptosis and necrosis. In this article, we observed evidence of necrotic type cell death such as electron-transparent cytoplasm and destroy of organelles. We did not observe electron-dense cytoplasm characteristic for apoptosis.
For representing the point clear for general readers, we rephrased corresponding sentence in the Results that now sounds as follow:
“Many cells looked electron-transparent, reflecting general degradation of cytoplasm and organelles that is characteristic for necrotic type cell death rather than apoptosis (Galluzzi et al., 2018).”
We also rephrased the legend for Figure 1 (new Figure 5) as follow:
“(A, B) Light and electron micrographs of the neuropil labeled with immunoperoxidase-DAB-Ni. Immunopositive mitochondria are pointed to with arrowheads. Notice that black immunoreaction end-product in (B) accumulates in the matrix while crista stay immunonegative.”
The author should provide the best representative Figure 1C1-C5. Images do not properly confirm significant SLP2 protein and CB1R-L31 antibodies complex accumulation in the mitochondrial matrix however, accumulation is higher at the cellular level.
Immunolabeling with nano-gold/silver (C1-C5) is much less sensitive than immunoperoxidase/DAB labeling. In this figure, labeling with gold/silver looks less intensive than amplifying DAB labeling. Same time, gold/silver provides precise antigen location in contact with inner mitochondrial membrane. We agree that more representative photos of the nano-gold/silver labeling could improve this figure. Unfortunately, time frame provided by the editor for corrections of the manuscript is not sufficient for replacing electron micrographs for better ones.
It would be nice if the author replaced the bar graph (Figure 4 A, B; Figure 5A, B) with a scatter graph. Since the error bar is larger it would be nice to see the individual value.
We tried to represent these data as scatter graphs. Unfortunately, very high numbers of analyzed mitochondria (Ns are in the range of hundreds) merge individual dots making the graphs less presentable. To fix this limitation, we included the estimation plots (new Figures 3C, 3D and 4C, 4D) that show frequencies of individual values equivalently to scatter graphs.
If the author found any changes in mitochondrial morphology between the hippocampus and cerebral cortex it would be nice if they were addressed in the discussion section. Representative images would be a plus.
We performed morphometric analyses only for mitochondria from the embryo neocortex. Yes, comparison of mitochondria from different brain segments would be informative. Unfortunately, applied 3D analysis is too time-consuming for this article. We are grateful for this suggestion that may be in the scope of our prospective studies.
Reviewer 3 Report
In the present work, the authors used TEM with 3D reconstruction to study the effects of anoxia on the Golgi apparatus (GA) and mitochondria in the embryonic mouse brain. After 1hr anoxia, they noted deformation of the Golgi apparatus. On the other hand, mitochondria seem to be affected after 3hr anoxia. Overall, the study is interesting and well-executed. Here are my suggestions:
1. What is Anoxia? How is it different from Hypoxia? I would suggest the authors discuss these issues briefly in the Introduction section.
2. Also, in line 28 of the Introduction, "reactive oxygen species neutralization" is described as one of the mitochondrial metabolic roles. Since mitochondria also contribute to generating ROS, I would suggest rephrasing it as "ROS homeostasis".
3. The discussion is short for this work. One interesting observation is the recovery of GA after reoxygenation followed by 1hr anoxia. However, there was no recovery of GA after reoxygenation followed by 3hr anoxia, when mitochondria are affected. How is the recovery of GA linked to healthy mitochondria? The authors should comment and speculate on this issue in a detailed way in the discussion section.
Author Response
Reviewer 3
In the present work, the authors used TEM with 3D reconstruction to study the effects of anoxia on the Golgi apparatus (GA) and mitochondria in the embryonic mouse brain. After 1hr anoxia, they noted deformation of the Golgi apparatus. On the other hand, mitochondria seem to be affected after 3hr anoxia. Overall, the study is interesting and well-executed. Here are my suggestions:
- What is Anoxia? How is it different from Hypoxia? I would suggest the authors discuss these issues briefly in the Introduction section.
We are grateful for this suggestion that makes the point clear. We added following sentence in the Introduction:
“Anoxic conditions assumed as total oxygen deprivation with excluded blood circulation were achieved through decapitation of embryos and placing the heads in artificial cerebro-spinal fluid (ACSF) devoid of dissolved oxygen.”
- Also, in line 28 of the Introduction, "reactive oxygen species neutralization" is described as one of the mitochondrial metabolic roles. Since mitochondria also contribute to generating ROS, I would suggest rephrasing it as "ROS homeostasis".
Corrected. We replaced word “neutralization” for "homeostasis".
- The discussion is short for this work. One interesting observation is the recovery of GA after reoxygenation followed by 1hr anoxia. However, there was no recovery of GA after reoxygenation followed by 3hr anoxia, when mitochondria are affected. How is the recovery of GA linked to healthy mitochondria? The authors should comment and speculate on this issue in a detailed way in the discussion section.
We understand the request for comprehensive discussion of this novel event. Unfortunately, we do not have an opportunity for dipper investigation of GA and mitochondria interactions in the frame of this article. We prefer to avoid long speculations without unequivocal experimental data that hardly will be appreciated by readership of respectful journal such as IJMS.
We added following sentences in Discussion:
“…if energy deficit continues long enough, GA malfunction may damage other organelles including mitochondria. In turn, damaged mitochondria may initiate numerous irreversible reactions such as reactive oxygen species production, calcium and cytochrome c leakage, cell apoptosis or necrosis, etc (Giacomello et al., 2020; Han et al., 2020; Harland et al., 2020). Thus, molecular architecture of mitochondria, at least in mouse embryo brain, is resistant to anoxia during about 1-hour period, whereas intracellular transportation and other functions executed by GA are vulnerable to shortage of ATP supply and likely participate in further cellular disorder.
To our knowledge, reciprocal influence of GA and mitochondria and, particularly, direct link between GA phenotype and ultrastructure of mitochondria have not been reported and deserve deep investigation. We do not know what molecular mechanisms are responsible for GA swirling. Which pathological conditions, besides acute anoxia as described in this paper, may be linked to up-regulation of onion-like GA? If identified, can GA malfunction help us understand the pathological processes? Can it serve for diagnosis or potential treatment of human diseases? Future experiments are warranted to address these and many other intriguing questions.”
Round 2
Reviewer 1 Report
I am satisfied with the authors' response to my questions. The manuscript could be accepted.
Author Response
Thank you for your positive decision. According to your suggestion, we had the manuscript redacted by a native English speaker.
Yury Morozov